# Mechanism of LolCDE as a molecular extruder of bacterial triacylated lipoproteins

Stuti Sharma [1,5], Ruoyu Zhou[2,5], Li Wan[2], Shan Feng[2], KangKang Song[3,4], Chen Xu[3,4], Yanyan Li [2✉] & Maofu Liao [1✉]

Lipoproteins are important for bacterial growth and antibiotic resistance. These proteins use lipid acyl chains attached to the N-terminal cysteine residue to anchor on the outer surface of cytoplasmic membrane. In Gram-negative bacteria, many lipoproteins are transported to the outer membrane (OM), a process dependent on the ATP-binding cassette (ABC) transporter LolCDE which extracts the OM-targeted lipoproteins from the cytoplasmic membrane. Lipid-anchored proteins pose a unique challenge for transport machinery as they have both hydrophobic lipid moieties and soluble protein component, and the underlying mechanism is poorly understood. Here we determined the cryo-EM structures of nanodisc-embedded LolCDE in the nucleotide-free and nucleotide-bound states at 3.8-Å and 3.5-Å resolution, respectively. The structural analyses, together with biochemical and mutagenesis studies, uncover how LolCDE recognizes its substrate by interacting with the lipid and N-terminal peptide moieties of the lipoprotein, and identify the amide-linked acyl chain as the key element for LolCDE interaction. Upon nucleotide binding, the transmembrane helices and the periplasmic domains of LolCDE undergo large-scale, asymmetric movements, resulting in extrusion of the captured lipoprotein. Comparison of LolCDE and MacB reveals the conserved mechanism of type VII ABC transporters and emphasizes the unique properties of LolCDE as a molecule extruder of triacylated lipoproteins.

[1] Department of Cell Biology, Blavatnik Institute, Harvard Medical School, Boston, MA, USA. [2] Key Laboratory of Structural Biology of Zhejiang Province, School of Life Sciences, Westlake University, Hangzhou, China. [3] Department of Biochemistry and Molecular Pharmacology, University of Massachusetts Medical School, Worcester, MA, USA. [4] Cryo-EM Core Facility, University of Massachusetts Medical School, Worcester, MA, USA. [5] These authors contributed equally: Stuti Sharma, Ruoyu Zhou. ✉email: liyanyan@westlake.edu.cn; maofu_liao@hms.harvard.edu

Found in all bacteria, lipoproteins are anchored in the cytoplasmic membranes using the lipid moiety attached to the N-terminal cysteine residue. Lipoproteins are a significant proportion (1–3%) of bacterial proteomes[1] and play central roles in bacterial physiology, including cell envelope formation, lipopolysaccharide biogenesis, nutrition acquisition, biofilm formation, stress response, and modulation of immune response of the host[2,3]. After being generated in the cytosol, lipoproteins are translocated across the membrane through SecYEG or twin-arginine systems, and subsequently acylated by sequential actions of a series of modification enzymes[4,5] (Fig. 1a). These enzymatic reactions result in tri-acylation of the N-terminal invariant cysteine residue in the lipoproteins, with two acyl chains ester-linked to the cysteine side chain and one acyl chain amide-linked to the N-terminus of lipoprotein. However, variable enzymes in different bacteria can lead to variation in the number and position of lipoprotein acylation[6,7].

In Gram-negative bacteria, many lipoproteins are transported to the outer membrane (OM), being positioned in the periplasmic leaflet to face the periplasm or in the outer leaflet to become surface exposed[8]. Lipoprotein transport from the inner membrane (IM) to the OM depends on a set of proteins in the Lol (localization of lipoprotein) pathway and is best studied in *E. coli*[9] (Fig. 1a). The OM-targeted lipoproteins are first extracted out of the IM, a process driven by the ATP-binding cassette (ABC) transporter complex LolCDE. All ABC transporters contain two transmembrane domains (TMDs) and two nucleotide-binding domains (NBDs). The transmembrane helices (TMs) of LolC and LolE form the TMDs, and two LolD proteins function as the NBDs. In addition, LolC and LolE each have a large periplasmic domain. The periplasmic chaperone LolA binds to the periplasmic domain of LolC, picks up extracted lipoprotein, and delivers it to the OM acceptor LolB which itself is a lipoprotein. Finally, the lipoprotein is inserted into the OM. While *E. coli* can grow without LolA and LolB under certain conditions, LolCDE is strictly essential for bacterial survival[10]. Due to its importance in OM localization of lipoproteins and antibiotic resistance, LolCDE is an attractive target for developing novel class of antibacterial drugs[11–13].

To fulfill its physiological function, LolCDE must not only specifically recognize mature lipoproteins, but also distinguish the lipoproteins that are destined to the OM from those which remain in the IM. The mechanism underlying these two fundamental aspects of LolCDE is poorly understood. In *E. coli*, the final acylation enzyme Lnt, which adds the third acyl chain to the N-terminal amine group, is an essential protein for bacterial growth, unless LolCDE is overexpressed[14]. Thus, N-acylation of lipoprotein likely serves as a checkpoint to ensure that LolCDE interacts with only mature lipoproteins. For the determinants of specific interaction between LolCDE and OM lipoproteins, our knowledge is limited. In *E. coli*, the presence of an aspartate residue in the second amino acid position (i.e., +2 position) immediately after the invariant cysteine residue, causes lipoprotein retention in the IM[9]. However, this well-known "+2 rule" is applicable to only *E. coli* and related enterobacteria. For other species, analyses based on primary amino acid sequence of the N-terminal lipoprotein residues fails to generate definitive patterns to predict the outcome of lipoprotein transport[15–18], suggesting that lipoprotein sequence alone is not sufficient for determining lipoprotein interaction with the transporter.

While numerous transporters that mediate cross-membrane movement of lipids or proteins have been extensively studied, lipoproteins containing both lipid and protein moieties represent a special group of substrates for membrane transport, and the underlying mechanism remains an enigma. The topological organization of LolCDE is distinct from that of most ABC transporters, and LolCDE is predicted to have similar folding as MacB, a homodimeric ABC transporter in a tripartite multidrug transporter complex in Gram-negative bacteria[19]. Based on TMD architecture, MacB is assigned as the only type VII ABC transporter with known structures[20,21]. Mainly due to the lack of a substrate-bound MacB structure, the mechanisms underlying substrate recognition and transport of this type of ABC transporters remain obscure. Importantly, unlike MacB which accepts a variety of compounds from the periplasm, LolCDE specifically extracts lipoproteins from the IM, and thus the functional mechanisms for these two ABC transporters must be divergent.

While the structures of LolCDE in detergent were reported recently[22], structural studies have not been performed for lipoprotein transporter in a native-like lipid bilayer. In this work, we determined the cryo-EM structures of nucleotide-free and ADP-vanadate-bound *E. coli* LolCDE in nanodiscs, at 3.8-Å and 3.5-Å resolutions, respectively. Structural analyses reveal the architecture of LolCDE, high-resolution details of lipoprotein-transporter interaction at the interface between LolC and LolE, and large-scale conformational transition induced by nucleotide binding. Together with biochemical assays and mutagenesis studies, our results reveal the fundamental mechanism of LolCDE by which the lipoprotein substrate is specifically captured and extracted from the membrane. Furthermore, the comparison between LolCDE and MacB provides important insights of how type VII ABC transporters function.

## Results

**Purification and structural determination of LolCDE in nanodiscs.** *E. coli* LolCDE complex was overexpressed in *E. coli* strain BL21(DE3), purified in dodecyl maltoside (DDM) detergent, reconstituted into nanodiscs with palmitoyl-oleoyl-phosphatidylglycerol (POPG), and screened in negative stain for monodisperse particles of similar size and shape (Supplementary Fig. 1a, b). The analysis of the kinetics of ATP hydrolysis by LolCDE in nanodiscs yielded a $Km$ value of $0.19 \pm 0.05$ mM and a $Vmax$ value of $242.8 \pm 15.3$ mole phosphate per min per mole protein (Fig. 1c). This activity is ~2.7 times the activity of LolCDE in DDM (Fig. 1b), suggesting that membrane environment is important to maintain the native conformation and full activity of the transporter. Because no lipoprotein was added in ATPase assay, the measured activities reflect the basal ATPase activity of LolCDE without active lipoprotein transport. LolCDE was sensitive to the inhibition by vanadate, and 0.1 mM orthovanadate led to ~90% suppression of the activity of LolCDE (Fig. 1b and Supplementary Fig. 1c). The nanodisc-embedded LolCDE was subjected to single-particle cryo-EM analysis, generating a cryo-EM three-dimensional (3D) reconstruction at an overall resolution of 3.8 Å (Fig. 1d and Supplementary Fig. 2). The TMs of LolC and LolE are with higher resolution and have well-defined side-chain densities for most amino acid residues, enabling de novo model building and unambiguous registry of amino acids (Supplementary Fig. 2g). LolD and the periplasmic domains of LolC and LolE are with lower resolution, likely due to higher mobility, and their models were built based on published domain structure and homology model as detailed in Methods.

**Overall structure of LolCDE.** The structure of *E. coli* LolCDE displays pseudo-two-fold symmetry (Fig. 1d), and the homologous LolC and LolE each interact with one LolD protein in the cytosol, using coupling helix between TM2 and TM3 and C-terminal sequence (Fig. 1g). As shown in the domain arrangement (Fig. 1e), LolC and LolE each contain an N-terminal elbow helix, four TMs (TM1-4), a large periplasmic

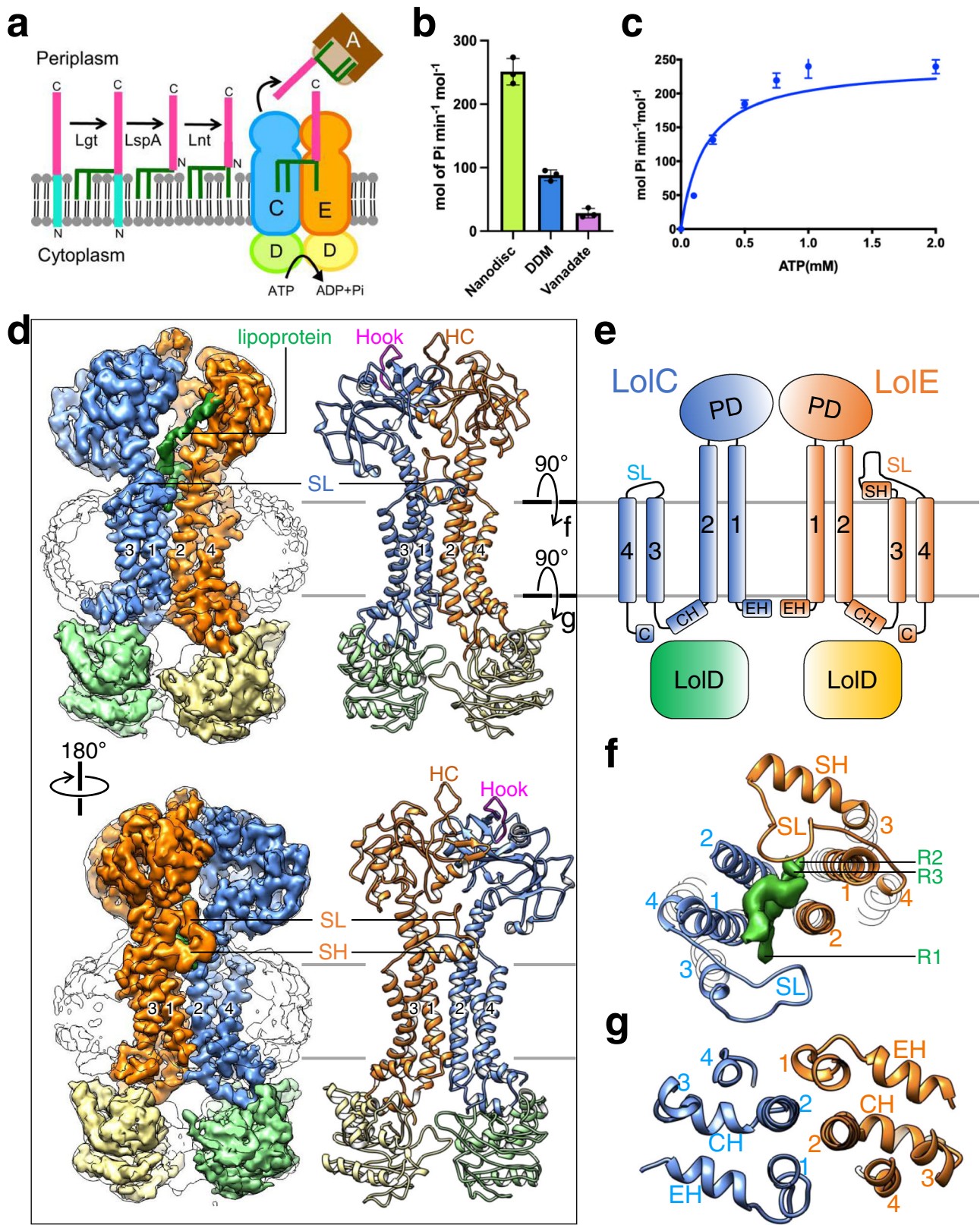

domain between TM1 and TM2, and a shoulder sequence between TM3 and TM4. In the center of LolCDE are TM1 and TM2, against which TM3 and TM4 are packed (Fig. 1f). The overall structure and TMD topology of LolCDE are similar to those of MacB, the founding member of type VII ABC transporters[19,20] (Supplementary Fig. 3).

While the structures of LolC and LolE, particularly in their TMDs, are highly homologous, with a root-mean-square-deviation (RMSD) of 1.12 Å over 145 Cα atoms of all TMs (Supplementary Fig. 3a–c), notable differences lie in their periplasmic domains and shoulder sequences. The periplasmic domains of LolC and LolE are oriented differently (Supplementary Fig. 3d). The β-hairpin loop in

**Fig. 1 Biochemical and cryo-EM studies of LolCDE. a** Diagram of the steps of lipoprotein biogenesis in the inner membrane, including three enzymatic processing steps, LolCDE-mediated extraction, and delivery to LolA. The Lol proteins are labeled with letters. **b** ATPase activity of LolCDE in nanodiscs, in DDM, and in nanodiscs with 0.1 mM vanadate. Shown is a representative of three independent experiments. Each point represents mean ± s.d. of three measurements in one experiment. **c** ATPase activities of LolCDE in nanodiscs. Each point represents mean ± s.d. of three separate measurements. **d** Cryo-EM map filtered at 3.8-Å resolution and model of LolCDE, viewed from the front (top) and back (bottom). LolC, LolE, two LolD subunits, and lipoprotein are colored separately. Nanodisc is shown as outline. The boundaries of the inner membrane are indicated by gray lines. Hook in LolC, Hook counterpart (HC) in LolE, shoulder loop (SL), and shoulder helix (SH) are indicated. Transmembrane helices are labeled with numbers. **e** Topology of LolCDE. C C-terminal sequence, CH coupling helix, EH N-terminal elbow helix. **f, g** Views perpendicular to the membrane plane of the cross sections indicated in (**d**).

LolC (termed the "Hook"), which binds to LolA for lipoprotein transport[23], is positioned sideway, while the Hook counterpart in LolE, not involved in LolA binding, is pointed upward (Fig. 1d). The shoulder sequence between the TM3 and TM4 of LolC has 18 amino acid residues (342–360) forming a loop structure. In comparison, the shoulder sequence of LolE is much longer and contains two parts: a 12-residue shoulder helix (344–356), which is located at the membrane surface, and a highly ordered 21-residue shoulder loop (357–378), which rises above the membrane surface by ~16 Å. As detailed below, the shoulder sequences play a crucial role in lipoprotein interaction.

**Three acyl chains of lipoprotein accommodated in two hydrophobic pockets.** A density of co-purified lipoprotein was clearly resolved in our cryo-EM structure of LolCDE, showing all three acyl chains sandwiched between LolC and LolE (Fig. 1d, f, and Fig. 2a–c). Notably, the three acyl chains attached to the invariant cysteine residue are lifted to the level of the membrane surface, adopting a nearly horizontal orientation (Fig. 2c, h). Thus, our structure captured an intermediate transport state of LolCDE, in which the lipoprotein substrate has been elevated to the interface between the IM and the periplasm but not yet released from the TMDs of the transporter.

The N-terminal cysteine residue and its three acyl chains are the conserved features of all mature *E. coli* lipoproteins, and thus expected to contribute to specific recognition of lipoproteins by LolCDE. This is consistent with the observation that the N-terminal peptide of lipoprotein and the acyl chains form intimate interactions with LolCDE and are well resolved in the cryo-EM map. The three acyl chains are separated into two groups and accommodated in two hydrophobic pockets (Fig. 3a, b). One of the two cysteine side chain-connected acyl chains (termed "R1") is located in the "front pocket" formed by the TM1 and shoulder loop of LolC and the TM2 of LolE (Fig. 2a, d, e and Supplementary Fig. 4f). The other side chain-connected acyl chain (termed "R2") and the N-terminal amine group-linked acyl chain (termed "R3") are packed together in the "back pocket" formed by the TM1 and shoulder loop of LolE and the TM2 of LolC (Fig. 2b, f, g and Supplementary Fig. 4g). Due to longer shoulder sequence in LolE and greater distance between the TMs, the back pocket accommodates a much larger volume than the front one. In the front pocket, R1 forms close contact with several hydrophobic residues from LolC (Val44, Val47, Met48 and Phe51 in TM1 and Leu351 in shoulder loop) and from LolE (M267 and Ile271 in TM2) (Fig. 2d, e). In the back pocket, the interactions are predominantly mediated between R3 and the shoulder loop of LolE, involving Phe360, Leu361, Ile365, Tyr366, Phe367, and Leu 371 (Fig. 2g). In comparison, R2 makes much less contact. Additional hydrophobic interactions with R3 and R2 are contributed by the TM2 of LolC (Met262, Met266, and Leu270) and the TM1 of LolE (Val43, Met48, and Phe51) (Fig. 2f). Most of the hydrophobic residues that form the substrate-binding pocket are highly conserved (Supplementary Fig. 5). Using single-site mutagenesis, the hydrophobic residues in close contact with the acyl chains of lipoprotein were changed to asparagine residue and

tested for their stable interaction with the OM lipoprotein Lpp by co-purification (Fig. 2j, k). On either side of R1, F51N of LolC (Fig. 2j) and M267N of LolE (Fig. 2k) did not block stable Lpp interaction. M266N of LolC in the back pocket also showed little effect in suppressing Lpp binding (Fig. 2j). In contrast, all single mutations in the shoulder loop of LolE (F360N, L361N, Y366N and L371N) abolished Lpp binding (Fig. 2k). These results indicate the critical importance of the LolE shoulder loop and back pocket in lipoprotein interaction. Interestingly, the above-mentioned single-residue mutants purified in DDM showed various ATPase activities, which did not correlate with their capability of stable Lpp binding (Supplementary Fig. 4i, j). Specifically, F267N, F360N, and L371N of LolE as well as F51N of LolC clearly decreased the activity of LolCDE, suggesting that, in addition to lipoprotein binding, these amino acids affect conformational transition of the transporter that is required for ATP binding and hydrolysis.

Our structure reveals that three acyl chains of lipoproteins occupy two hydrophobic pockets in LolCDE with the last attached acyl chain (R3) making the most extensive interactions. These observations explain the preference of LolCDE for mature, triacylated lipoprotein, as well as the differential functional requirement of individual acyl chains in lipoprotein for transport. In *E. coli*, deletion of Lnt, and thus removal of R3 from lipoproteins (Fig. 1a), is lethal, and can be rescued by over-expression of LolCDE[14]. Interestingly, the lack of Lnt can also be complemented by expressing a transacylase, Lit, which transfers one of the two side chain-linked acyl chains to the amine group (R3)[24]. These findings corroborate the notion that, among the three acyl chains in lipoprotein, R3 in the back pocket forms the strongest interaction with LolCDE and is crucial for lipoprotein transport.

**Interaction of N-terminal peptide of lipoprotein with LolCDE.** In our cryo-EM map of LolCDE, a strong density is resolved following the triacylated cysteine residue, corresponding to 6 N-terminal amino acid residues of lipoprotein. This density is not with sufficient resolution for accurate amino acid assignment, likely due to different lipoproteins bound to the purified LolCDE. Lpp is the most abundant lipoprotein in *E. coli*[25,26], and can form stable complex in our co-purification assay (Fig. 2j, k). Indeed, mass spectrometric analysis of purified LolCDE revealed various OM lipoproteins including Lpp (Supplementary Fig. 1d). Because Lpp and many other OM lipoproteins contain a serine residue in the +2 position, we modeled the first six residues of the lipo-protein as Cys-Ser-Ala-Ala-Ala-Ala.

The first four residues (+1 to +4 positions) are at approximately the same height above the membrane surface. They adopt a kinked conformation to tightly fit into a relatively shallow hydrophobic pocket (Fig. 3b, best shown in the fourth panel), which is formed by the TM1 and TM2 of LolC and the shoulder loop and TM2 of LolE (Fig. 2i and Supplementary Fig. 4h). While the +1 cysteine residue is in closer proximity to the TM2 of LolE, the +2 residue is rotated away and positioned near Ile365 and Tyr366 (shoulder loop) of LolE. The following

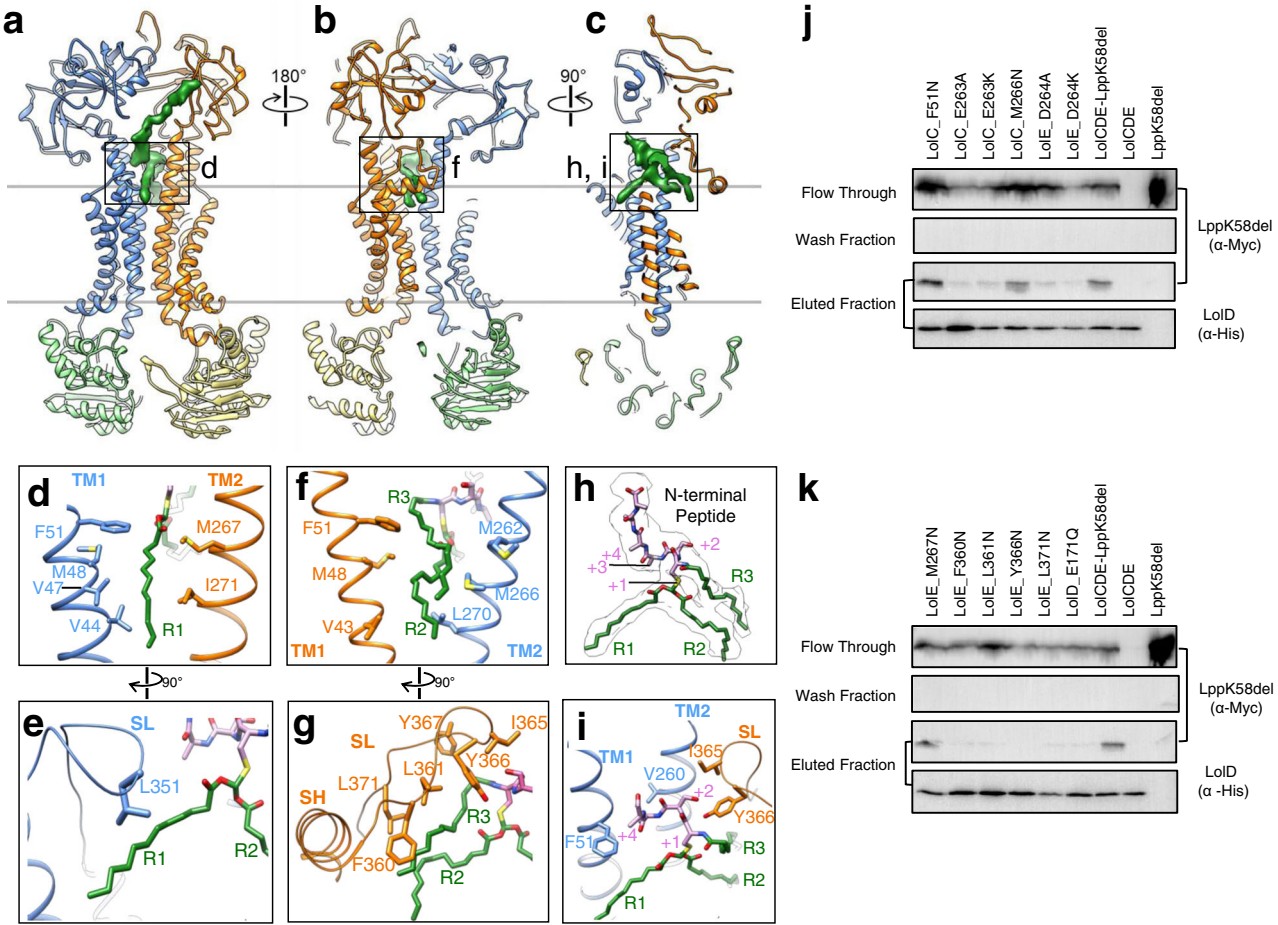

**Fig. 2 Lipoprotein interaction with LolCDE. a–c** Model of LolCDE with the cryo-EM density of lipoprotein (green), shown as front (**a**), back (**b**), and sectional side (**c**) views. Lol proteins are colored as in Fig. 1. **d–i** Close-up views of the selected areas indicated in (**a–c**). The lipoprotein is shown with the N-terminal peptide in purple and three acyl chains in green: the N-terminal R3 and cysteine side chain-connected R1 and R2. SH shoulder helix (only in LolE), SL shoulder loop (in LolC and LolE). **j, k** Co-purification assay to test Lpp binding to wild-type and mutant LolCDE. LppK58del without the C-terminal lysine residue was used to prevent cell toxicity. Except the last two lanes, where only wild-type LolCDE or LppK58del was individually expressed, all lanes show the results from co-expression of LolCDE and LppK58del. LolCDE was purified using Ni-NTA via the His tag on LolD. Flow-through, wash, and elute were analyzed by western blot. Data are representative of three independent experiments.

N-terminal peptide takes a right-handed turn such that the +3 and +4 residues come in close contact with Val260 (TM2) and Phe51 (TM1) of LolC, respectively. The following ~10 residues extend upward and reach the periplasmic domain of LolE, where it makes contact with a patch of hydrophobic residues consisting of Tyr248, Val249, Tyr250, and Ile251 (Supplementary Fig. 4d). The observed N-terminal sequence of lipoprotein reaching to the periplasmic domain of LolE may contribute to optimal lipoprotein processing by LolCDE. A recent study revealed that half of the OM lipoproteins in *E. coli* have a long and intrinsically disordered N-terminal sequence which is important for efficient lipoprotein transport to the OM[27]. The further C-terminal region of lipoprotein is not resolved, suggesting that the main body of lipoproteins is located outside the periplasmic domains and does not interact with LolCDE. These are consistent with the notion that LolCDE recognizes highly variable lipoproteins through specific interactions with only their N-terminal peptide and lipid moieties. Furthermore, localization of N-terminal region of lipoprotein on the front side of LolCDE, together with the large shoulder sequence of LolE shielding the back side, seems to suggest that lipoproteins enter the transporter from the front side through the interface between the TM1 of LolC and the TM2 of LolE.

Above the acyl chain binding pockets, the surface property at the interface of LolC and LolE transitions sharply from hydrophobic to hydrophilic with predominantly negative charge (Fig. 3c, d). The N-terminal cysteine residue is sandwiched between Glu263 in LolC and Asp264 in LolE (Supplementary Fig. 4b, c). Changing these two residues individually to alanine or lysine residue blocked stable attachment of Lpp to LolCDE (Fig. 2j), demonstrating the importance of negative charge for lipoprotein interaction. Interestingly, the mutations of Glu263 in LolC demonstrated much stronger inhibitory effect on the ATPase activity of LolCDE (Supplementary Fig. 4j). Additional negatively charged residues include Asp352 (shoulder loop of LolC), Glu54 (LolE-TM1), and Asp364 (shoulder loop of LolE) (Fig. 3d). While the mechanism by which the negatively charged surface supports lipoprotein binding is not clear, charge repulsion with the phosphate groups of the phospholipids in the IM may limit their entry to LolCDE and help select for lipoproteins with an uncharged cysteine residue at the +1 position.

**LolCDE extrudes lipoprotein via drastic conformational transition.** To understand how LolCDE extracts the bound lipoprotein out of the TMDs, we determined the cryo-EM structure of vanadate-trapped *E. coli* LolCDE at 3.5-Å resolution (Fig. 4a and

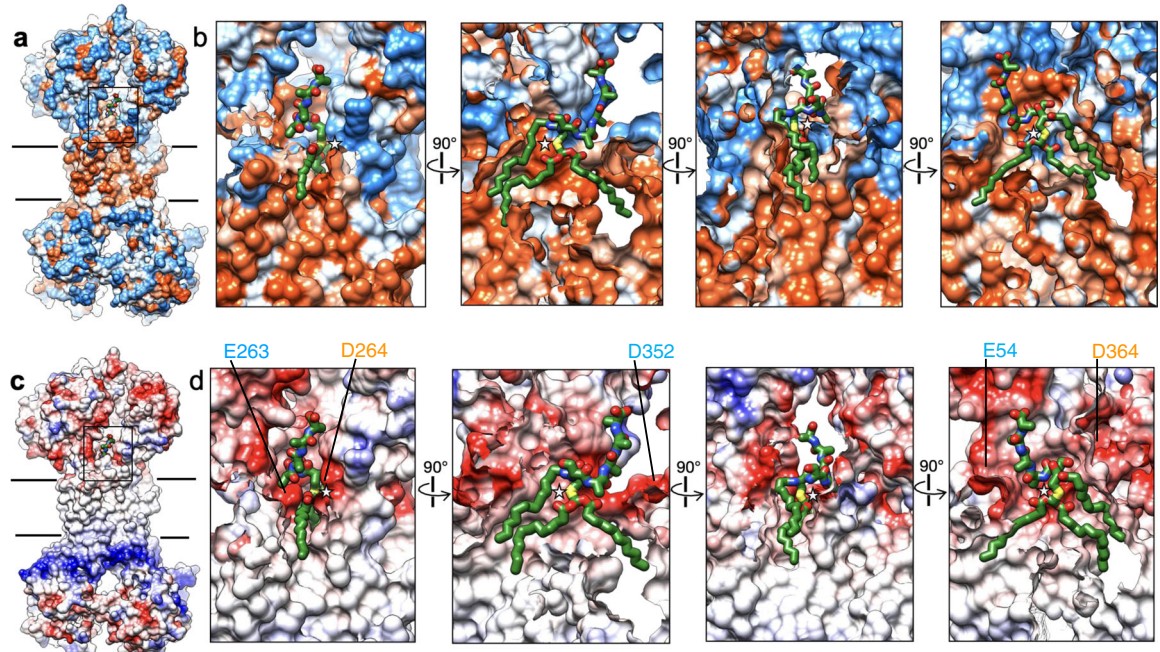

**Fig. 3 Hydrophobic and electrostatic surface properties of lipoprotein binding site. a** Hydrophobic surface representation of LolCDE, showing hydrophobic (orange) and hydrophilic (blue) areas. **b** Close-up sectional views from 4 orthogonal angles of the area indicated in (**a**). **c** Electrostatic surface representation of LolCDE, showing areas of positive (blue) and negative (red) charge. **d** Close-up sectional views from 4 orthogonal angles of the area indicated in (**c**). Lipoprotein is shown as green stick. The cysteine residue at +1 position is marked with a star. Negatively charged residues near the lipoprotein are indicated.

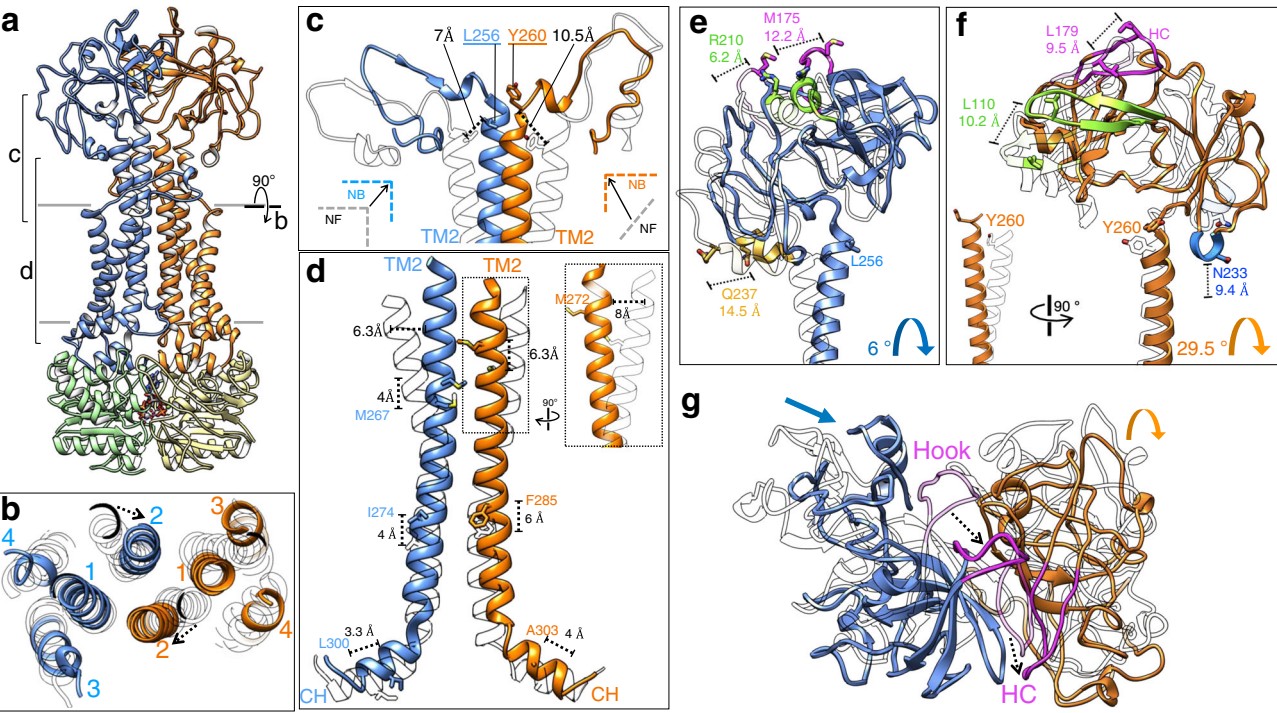

**Fig. 4 Conformational changes in LolCDE upon vanadate trapping. a** Cryo-EM structure of vanadate-trapped LolCDE with subunits colored as in Fig. 1. **b** Cross-sectional view perpendicular to the membrane plane indicated in (**a**). **c** T-shaped structure formed by the 90° kinks of both TM2 helices. Displacement of the Cα atoms of selected residues is shown. **d** Propagation of upward movement from coupling helix (CH) to TM2, measured as displacement of Cα atoms of indicated residues. Towards the N-terminal region, both TM2 helices undergo inward movement. The more pronounced inward shift of the TM2 in LolE is highlighted in dotted boxes. **e** Translation and rotation of the periplasmic domain of LolC is illustrated by comparison of nucleotide-free (gray) and vanadate-trapped (color) LolCDE. Selected structural regions are highlighted and the displacement of Cα atoms of indicated residues shown. **f** Same as (**e**), except for the periplasmic domain of LolE. **g** Top-down view of periplasmic domains showing overall conformational changes. Hook in LolC and the hook counterpart (HC) in LolE are colored magenta.

Supplementary Fig. 6). The cryo-EM sample contained 1 mM vanadate, a concentration sufficient for ~90% inhibition of LolCDE (Supplementary Fig. 1c). Such inhibition is caused by ADP-vanadate complex trapped in ATP site and stabilizing an intermediate conformation of ATPase.

In the vanadate-trapped LolCDE structure, the two LolD proteins move towards each other and associate together, with one ADP-vanadate complex in each of the two ATP sites at the dimer interface (Supplementary Fig. 7d). LolD engages three regions of both LolC and LolE: coupling helix, C-terminal sequence, and elbow helix. Among these three structural elements, coupling helices display the most pronounced conformational shift, which in turn causes an upward and inward movement of TM2 helices (Fig. 4b, d and Supplementary Movie 1). The conformational changes in TM2 and coupling helix of LolC and LolE are asymmetric, with more movements observed in LolE than LolC. Specifically, the coupling helix in LolC pushes TM2 up by ~4 Å, and such movement in LolE leads to a remarkable upward shift of TM2 by ~6 Å, which is more than one helical turn (Fig. 4d). In addition to vertical movement, the TM2 helices of LolC and LolE also shift inward at the membrane-periplasm interface by ~6.3 and ~8 Å, respectively (Fig. 4b, d). In sharp contrast to the substantial conformational transition of TM2, all other TMs demonstrate little movement, with the exception of the TM1 of LolE, which displays an inward shift by ~6 Å following the TM2 of LolE but no upward movement (Fig. 4b). In nucleotide-free LolCDE, the lipoprotein bound between the two TM2 helices is already elevated to the level of membrane surface (Fig. 2c). Upon nucleotide binding, the upward shift of TM2 would push the lipoprotein out of the TMDs and into the space between the periplasmic domains. Therefore, the power stroke is initiated when two NBDs (i.e., LolD) dimerize upon ATP binding, and, via the TM2 helices across the membrane, the movement of NBDs in the cytosol is coupled to the extrusion of the bound lipoprotein in the periplasm.

The inward movement of TM2 helices results in a two-helix bundle in the center of the transporter, thereby collapsing all three pockets which, in the nucleotide-free conformation, accommodate three acyl chains and the N-terminal peptide of the bound lipoprotein. Accordingly, we do not observe lipoprotein in the structure of vanadate-trapped LolCDE. When the catalytically important Glu171 in LolD was mutated to glutamine residue (E171Q), LolCDE lost stable Lpp binding (Fig. 2k). This is consistent with the notion that the mutant LolCDE in the IM is stabilized in the ATP-bound conformation which cannot bind lipoprotein. Notably, TM2 closes the lateral opening between LolC and LolE in the OM leaflet and blocks access of both shoulder loops to the substrate (Supplementary Fig. 7c, e), and the resulting destabilization of the shoulder loops is clearly manifested by the weaker cryo-EM density in the vanadate-trapped conformation (Supplementary Fig. 7f). Complete elimination of all lipoprotein accommodating pockets and closure of substrate entry gate may be important to prevent lipoprotein backloading and ensure transport completion before accepting a new substrate. Taken together, our cryo-EM structure of vanadate-trapped LolCDE represents a functional state after the extrusion of lipoprotein.

**Large-scale movements of LolCDE in the periplasmic space.** The drastic shifts of TM2 propagate into the periplasmic regions of LolCDE. In the position of Tyr260 at the periplasmic end of TM2 of LolE, ~10 Å above the membrane surface, a local structural rearrangement creates a 90° kink, which travels 10.5-Å distance upward and to the center (Fig. 4c). A corresponding

90° kink in LolC is present in the nucleotide-free conformation, and, upon vanadate trapping, also moves upward and inward to join the 90° kink from LolE. These two kinks together form a T-shaped structure, closing the membrane-proximal space and shrinking the opening on the front side of the inter-periplasmic domain space (Supplementary Fig. 7c).

The upward and inward movements of TM2 cause remarkable translocation and rotation of periplasmic domains, resulting in an overall closure of these two domains. The asymmetrical transitions of the two TM2 helices are also reflected in distinct movements of the two periplasmic domains. While the motion of LolC periplasmic domain is predominantly a translation toward the center by ~12 Å with a small rotation of ~6° (Fig. 4e), the periplasmic domain of LolE moves mainly by a rotation of ~29.5° (Fig. 4f and Supplementary Movie 1). The motion in LolC relocates the LolA-binding Hook near the top of the transporter (Fig. 4e). The crystal structure of LolA in complex with the periplasmic domain of LolC[23] can be docked on our nucleotide-free and vanadate-trapped LolCDE structures without obvious clashes (Supplementary Fig. 8), which is consistent with the previous findings that LolA-LolCDE interaction is not affected by the nucleotide binding to LolCDE[23]. Thus, the release of LolA-lipoprotein complex from LolC is likely caused by the conformational change of LolA upon lipoprotein loading, rather than the periplasmic domain movements alone. In addition, the lipoprotein linker-binding loop in the periplasmic domain of LolE moves diagonally upward by ~8 Å, likely facilitating lipoprotein extraction by directly pulling the N-terminal peptide (Supplementary Fig. 4e). Together, the structural rearrangements of LolC and LolE in the periplasmic space seem to help lipoprotein extrusion, protect the acyl chains of extracted lipoprotein from the aqueous environment of the periplasm, and reorient the lipoprotein for interaction with the LolC-bound LolA.

## Discussion

Bacterial lipoproteins represent a special group of macromolecules for cross-membrane translocation, due to their amphipathic nature. Our results provide insights of how lipoproteins are specifically transported by a dedicated molecular machine. We propose a model of lipoprotein extraction by *E. coli* LolCDE (Fig. 5) in which (1) lipoprotein in the periplasmic leaflet of the IM laterally enters the transporter through the interface between LolC and LolE on the front side; (2) the acyl chains of lipoproteins form extensive hydrophobic interactions with the front and back pockets in the transporter, resulting in elevation of the lipoprotein to the level of membrane surface; (3) ATP binding-induced LolD dimerization causes TM2 to move upward, which in turn pushes the bound lipoprotein out of the TMDs and into the space between the two rearranged periplasmic domains. Finally, ATP hydrolysis leads to dissociation of NBD dimer, resetting the conformation of LolCDE for the next cycle of transport.

Our results reveal how LolCDE recognizes highly variable OM-targeted lipoproteins through three distinct pockets that interact with the conserved structural features in lipoproteins: acyl chains and the N-terminal peptide. Importantly, we identified the N-terminal amine group-linked acyl chain (R3) as the predominant element to mediate hydrophobic interaction with the transporter, which explains why LolCDE prefers fully mature lipoproteins after the final acylation step. The first four residues of lipoprotein form a kink to fit tightly in a hydrophobic pocket with a front opening, suggesting that the N-terminal peptide must adopt a structure with complementary properties with respect to the pocket. Thus, whether a lipoprotein is captured or avoided by LolCDE is reflected upon the combinatory effect of the sequence

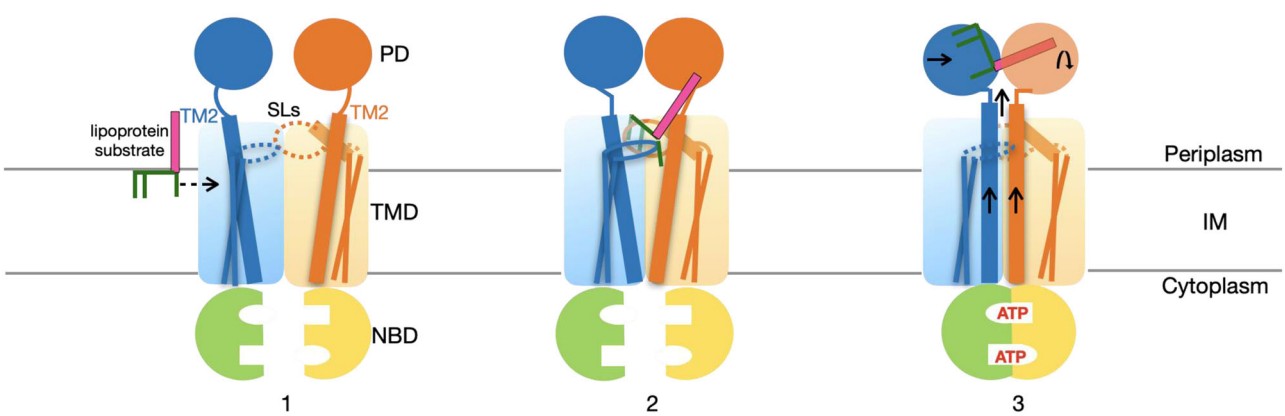

**Fig. 5 Proposed model for LolCDE-driven lipoprotein extraction from the inner membrane of *E. Coli*.** Lol proteins and lipoproteins are colored as in Fig. 1. See text for description of proposed steps for lipoprotein transport. SL shoulder loop (dotted oval), PD periplasmic domain, TMD transmembrane domain, NBD nucleotide-binding domain, IM inner membrane.

and folding of the N-terminal peptide as well as the shape and surface property of the peptide-accommodating pocket in the transporter. This likely contributes to the highly divergent Lol avoidance signals in different bacteria.

LolCDE is an attractive target for developing novel class of antibiotics. Three types of small molecules have been identified to target LolCDE, including pyridineimidazole[13], pyrazole[12], and pyrrolopyrimidinedione[11] compounds. The mode of action of these compounds is not well understood and resistant mutations occur at high frequency, both limiting further development of LolCDE inhibitors. For all three types of compounds, resistant mutations are predominantly mapped to two regions: the shoulder loop of LolE and the periplasmic portion of TM2 of LolC. Additional mutations are in other regions of LolE: shoulder helix and the periplasmic region of TM1. Only a few mutations are in the periplasmic domains. Together, these mutations suggest that resistance to LolCDE inhibitors is achieved by altering the lipoprotein binding pockets and affecting the conformation in the periplasmic region of LolCDE.

Recently Tang et al. published various cryo-EM structures of *E. coli* LolCDE in detergent (lauryl maltose neopentyl glycol, LMNG) using four different conditions: no added nucleotide, β-γ-imidoadenosine 5′-phosphate (AMP-PNP), continuous ATP turnover with LolA, and ADP[22]. Their lipoprotein-bound LolCDE structures (with or without LolA binding) from all four conditions display essentially identical conformation, and the four different structures (apo, lipoprotein-bound, lipoprotein- and LolA-bound, and NBD-closed) were proposed to represent four states in lipoprotein transport[22]. Our structure of nucleotide-free LolCDE in nanodiscs is similar to the lipoprotein-bound LolCDE structures from Tang et al., but with a critical difference in the conformation of R3 of the lipoprotein. While R3 is pointing down and extending away from the LolE shoulder loop in the structure from Tang et al., in our structure R3 adopts a nearly horizontal orientation forming extensive interaction with the entire central sequence of the LolE shoulder loop (amino acids 360–371) (Fig. 2g, k). This interaction appears to be the most extensive hydrophobic contact between lipoprotein acyl chains and LolCDE, thus identifying R3 and the LolE shoulder loop as the key recognition elements from the substrate and transporter, respectively. Compared to the AMP-PNP-bound, NBD-closed LolCDE structure at 4.1-Å resolution from Tang et al., our structure of vanadate-trapped LolCDE at 3.5-Å resolution shows better resolved side-chain densities in TMDs and more defined periplasmic domains and shoulder sequences. In our structure, the periplasmic regions of the TM2 helices and periplasmic

domains are positioned more towards the center resulting in a more compact conformation. The observed difference may have important functional implication, because a previous study shows that LolCDE in detergent can release the bound lipoprotein upon the treatment of ATP or vanadate, but not AMP-PNP[28]. Furthermore, the high-quality EM density of our vanadate-trapped LolCDE map allowed for accurate analysis of conformational transition of individual TM helices upon nucleotide binding (Fig. 4), which further enabled the comparison of the conformational transitions of LolCDE and MacB to derive the common mechanism of type VII ABC transporters (see below). Tang et al. obtained another structure of apo LolCDE, in which LolC and LolE open in a scissor-like motion, drastically increasing the distances between the two periplasmic domains and between the two NBDs. Interestingly, we did not observe such conformation throughout our cryo-EM analyses of the nucleotide-free and vanadate-trapped LolCDE in nanodiscs. Finally, compared to the ATPase activity of LolCDE in LMNG, the activities of LolCDE in nanodiscs and in DDM (Fig. 1b) are over five and two times higher, respectively. Consistent with its lower activity, LolCDE in LMNG demonstrates only half conversion to NBD-closed conformation upon AMP-PNP inhibition, which is in contrast with full NBD-closure of vanadate-trapped LolCDE in nanodiscs (Supplementary Fig. 6c), and a major portion of LolCDE in LMNG does not release the bound lipoprotein under continuous ATP hydrolysis condition (lipoprotein- and LolA-bound LolCDE structure)[22]. Taken together, most differences between our LolCDE structures and those from Tang et al. likely stem from the distinct environments in which LolCDE resides.

The structures of MacB in different functional states were resolved, and demonstrate how ATP binding causes large-scale conformational changes of the periplasmic domain, presumably harnessing mechanotransmission to drive the movement of MacA in the periplasm and expel the drug substrates[29–32]. The observed conformational change of LolCDE upon vanadate trapping is similar to that of MacB between its nucleotide-free and ATP-bound states[29,32] (Supplementary Movies 1, 2). In both cases, NBD dimerization causes the upward and inward movements of TM2 and subsequent rearrangement of periplasmic domains, which appear to be the common and distinct features of type VII ABC transporters. However, there are several important differences between MacB and LolCDE. First, while all 8 TMs of MacB move inward for its "bellows-like" function, with two TM1s and two TM2s forming a four-helix bundle in the center (Supplementary Fig. 9a, b), the inward movement in LolCDE is

limited to 3 TMs, with two TM2s forming a central two-helix bundle (Fig. 4b). Second, while the rotation of MacB periplasmic domain is facilitated by upward TM2 motion against inward TM1 motion, corresponding rotation in LolE is largely facilitated by the formation of the 90° kink at the periplasmic end of TM2 (Fig. 4c). Third, while MacB moves both TM1 and TM2 to close the periplasmic domains through symmetrical translation and rotation (Supplementary Fig. 9c, d), LolCDE predominantly uses the upward TM2 motion to extrude the bound lipoprotein and produce an asymmetrical movement of the two periplasmic domains which may have additional functions in protecting and reorienting the extracted lipoprotein to facilitate its loading onto LolA. In summary, all these differences highlight the distinct functions of LolCDE and MacB as membrane extractors and mechanotransducer, respectively.

## Methods

**Cloning, expression, and purification of LolCDE**. The genes encoding LolC, LolE, and LolD with flanking restriction sites BamHI and NotI were amplified individually from *E. coli* K12 genomic DNA by PCR. The C-terminus of LolD was extended using the linker sequence GGGAA and a 6x His tag. LolC, LolE, and LolD amplicons were individually inserted into pQlinkN vectors (Addgene). The recombinant vectors pQlinkN-LolD-His, pQlink-LolE and pQlink-LolC were individually digested using the restriction enzymes PacI and SwaI, and the digested products were linked using ligation independent cloning[33]. Site-directed mutants were constructed with the Q5 site-directed mutagenesis kit (NEB) according to the manufacturer's protocols (mutagenesis primers listed in Supplementary Table 2). The final recombinant vector pQLink-LolCD(6xHis)E was transformed into *E. coli* BL21(DE3) cells for over-expression. Transformed *E. coli* was grown in terrific broth supplemented with 100 μg/ml ampicillin at 37 °C until the cells reached an $OD_{600}$ of ~2. LolCDE expression was induced by the addition of 0.1 mM Isopropyl β-D-1-thiogalactopyranoside (IPTG) and the cells were grown at 18 °C for 48 h. Cells were collected by centrifugation and the pellets resuspended in buffer A (25 mM Tris pH 7.4, 250 mM NaCl and 10% glycerol). Resuspended cell pellets were flash frozen in liquid nitrogen and stored at −80 °C until use. Thawed cells were supplemented with 0.5 mg/ml lysozyme, 0.1 mg/ml DNase I, incubated on ice for 30 min, supplemented with protease inhibitors, and lysed by passing through an LM20 microfluidizer (Microfluidics) once. Lysed cells were subjected to low-speed centrifugation at $12,000 \times g$ for 30 min to remove unbroken cells and debris followed by ultracentrifugation at $100,000 \times g$ for 1 h to collect membranes. Membrane pellets were resuspended in buffer A, supplemented with protease inhibitors, and solubilized in 1% n-Dodecyl-β-D-Maltoside (Anatrace), for 1 h at 4 °C. Unsolubilized material was removed by ultracentrifugation at $100,000 \times g$ for 1 h. Solubilized membranes were subjected to affinity column chromatography using Ni-IMAC resin (Profinity) and eluted using buffer containing 25 mM Tris pH 7.4, 150 mM NaCl, 0.05% DDM, and 250 mM Imidazole. The eluted protein was further purified by size-exclusion chromatography on a Superdex 200 column in a buffer containing 25 mM Tris, pH7.4, 150 mM NaCl, 0.05% DDM, and 5% glycerol.

**Mass spectrometry identification of LolCDE and lipoproteins**. The purified LolCDE with bound lipoproteins was digested to peptides by trypsin for mass spectrometry identification. The peptides were separated by a home-made fused silica capillary column (75 μm internal diameter, 150 mm length; Upchurch, Oak Harbor, WA) packed with C-18 resin (300 Å, 5 μm, Varian, Lexington, MA) and performed using Thermo EASY1200 integrated nano-HPLC system. The MS data were collected in the data-dependent acquisition mode using Thermo QE HF-X mass spectrometer (Thermo Fisher Scientific). The single full-scan mass spectrum in the Orbitrap (400–1800 *m/z*, 60,000 resolution) was generated by 20 data-dependent MS/MS scans in the Orbitrap (100–1500 *m/z*, 15,000 resolution) using 30% normalized collision energy. Each mass spectrum was analyzed using the Thermo Xcalibur Qual Browser and screened in *E. coli* database using Proteome Discoverer.

**Co-purification of LolCDE and Lpp**. Gene encoding *lpp* was amplified from *E. coli* K-12 genomic DNA. The C-terminal lysine residue (K58) was deleted (LppK58del) to inhibit the formation of a covalent linkage with peptidoglycan and prevent cell toxicity. The fragment of *lpp* with C-terminal Myc tag was inserted into pQLink-LolCDE vector. All site-directed mutations were generated following the protocol of NEB Q5 site-directed mutagenesis kit. The final recombinant vector pQLink-LolCD(6xHis)E-Lpp (C-Myc) was transformed into *E. coli* BL21(DE3) cells for co-expression. The LolCDE–Lpp or relevant mutant proteins were expressed and purified using the same methods as described for LolCDE above. His tag purified LolCDE-Lpp fractions were detected by western blot using anti-His and anti-Myc antibodies.

**Nanodisc reconstitution**. 1-palmitoyl-2-oleoyl-sn-glycero-3-phosphatidylglycerol (POPG) (Avanti Polar Lipids) in chloroform was dried under argon gas and stored in vacuum overnight. The dried lipid film was re-suspended in nanodisc buffer (25 mM Tris pH 7.4, 150 mM NaCl), sonicated in a water bath for 1 h (until homogeneous) and solubilized in 25 mM sodium cholate. MSP1D1 membrane scaffold protein and purified LolCDE were added to the reconstitution mixture at a final molar ratio of 1:2:130 (LolCDE: MSP1D1: POPG) and incubated at 4 °C for 1 h. Detergent was removed by incubation with 0.6 g/ml Bio-Beads SM-2 (Bio-Rad) at 4 °C for 2 h. Nanodisc-reconstituted LolCDE was further purified by size exclusion chromatography using a Superdex 200 column in nanodisc buffer. The purity of LolCDE in nanodiscs was assessed using SDS-PAGE and negative stain electron microscopy.

**ATPase assay**. ATPase activity of LolCDE in DDM detergent or nanodiscs was measured using a colorimetric ATPase kit (Sigma Aldrich) according to the manufacturer's instructions. Briefly, 1 μg of LolCDE was incubated in 25 mM Tris, pH 8, 150 mM NaCl, 4 mM ATP, and 4 mM $MgCl_2$ for 30 min at 37 °C. The reaction was stopped by the addition of 200 μl of reagent provided in the kit, incubated at room temperature for 30 min and the absorbance at 620 nm was measured using a SpectraMax M5 spectrophotometer (Molecular Devices). Phosphate standard curve was constructed using stock solutions provided in the kit according to the manufacturer's instructions and used to determine the total concentration of released phosphate. ATPase activities of all samples were determined using the mean value of the samples according to the linear regression of standards. Data were plotted and analyzed in GraphPad Prism 8.

**Electron microscopy sample preparation and data acquisition**. Nanodisc-embedded LolCDE at a concentration of 1.6–2 mg/mL was used for freezing cryo-EM grids. A 2.5 μL volume of sample was applied to glow-discharged Quantifoil R1.2/1.3 holey carbon grids and blotted for 3.5 s at 100% humidity using a Mark IV Vitrobot (Thermo Fisher Scientific) before being plunge frozen in liquid ethane cooled by liquid nitrogen. For vanadate trapping, the samples were incubated in a buffer containing 2 mM ATP, 2 mM $MgCl_2$, and 1 mM sodium orthovanadate for 30 min at room temperature before applying the samples to cryo-EM grids. Cryo-EM images were collected at liquid nitrogen temperature on a Titan Krios (Thermo Fisher Scientific) equipped with a K3 detector (Gatan) and a BioQuantum imaging filter, using image shift and beam tilt to collect one shot per hole and nine holes per stage move. Movies were recorded in super-resolution mode with SerialEM[34] or AutoEMation[35]. A slit width of 20 eV for the energy filter was set during the data collection. The details of EM data collection parameters are listed in Supplementary Table 1.

**Electron microscopy image processing**. EM data were processed as previously described[36] with minor modifications. For both negative-stain EM and cryo-EM, particle images were initially selected using SamViewer (v21.01) and a semi-automated procedure implemented in Simplified Application Managing Utilities for EM Labs (SAMUEL v21.01)[37], and two-dimensional (2D) classification of selected particle images was performed with "samclasscas.py", "samtree2dv3.py" or 2D classification in RELION-3.0[38]. For processing cryo-EM images, dose-fractionated super-resolution movies were binned over $2 \times 2$ pixels, and beam-induced motion was corrected using the program MotionCor2[39]. Defocus values were calculated using the program CTFFIND4[40]. Initial models for 3D classification were generated by refinement of 2D class averages against a random density using projection matching. 3D classification and refinement were carried out in RELION-3.0. Following two rounds of global 3D classification, masks were constructed to focus 3D classification on LolCDE, omitting the signal from nanodisc. The orientation parameters of the homogenous set of particle images in selected 3D classes were iteratively refined to yield higher resolution maps using the "auto-refine" procedure in RELION. All refinements followed the gold-standard procedure, in which two half datasets are refined independently. The overall resolutions were estimated based on the gold-standard Fourier shell correlation (FSC) = 0.143 criteria. Local resolution variations were estimated from the two half data maps using ResMap[41]. The final map of nucleotide-free LolCDE was subjected to a density-modification procedure[42]. The final map of vanadate-trapped LolCDE was corrected for amplitude information by using "relion_postprocess" in RELION3.0. The detailed workflows of processing the cryo-EM datasets are illustrated in Supplementary Figs. 2c and 6c. The number of particles in each dataset and other details related to data processing are summarized in Supplementary Table 1.

**Model Building and refinement**. The crystal structure of MacB with ATPγS bound (PDB 5LIL [https://doi.org/10.2210/pdb5LIL/pdb]) was used as a template to generate homology models for LolC and LolE using SWISS-MODEL[43]. The homology models were fit into the cryo-EM maps for LolCDE in the nucleotide-free and nucleotide-bound conformations using UCSF Chimera[44]. Manual adjustment of the models was performed in COOT[45], followed by iterative rounds of real space refinement in PHENIX[46] and manual adjustment in COOT. The crystal structure of the periplasmic domain of LolC (PDB 6F3Z [https://doi.org/10.2210/pdb6F3Z/pdb]) was docked into the corresponding density in our maps, manually adjusted in COOT and real space refined in PHENIX. The refined model

for the periplasmic domain of LolC was used as a template to generate a homology model for the periplasmic domain of LolE using SWISS-MODEL. Similarly, a template-independent homology model was generated for LolD. The homology models were fit into their corresponding densities in UCSF Chimera, manually adjusted in COOT and real space refined in PHENIX. A SMILES string for the triacyl-peptide ligand was generated using the PubChem draw structure tool and restraints for the molecule were generated using PHENIX eLBOW[47]. For the ADP-Vanadate complex, the pdb three-letter-code (AOV) was used to generate restraints in eLBOW. The ligands were roughly fit into their corresponding densities in UCSF Chimera, manually adjusted with the CIF restraints in COOT and real space refined in PHENIX.

**Map visualization and structure analysis**. Maps were visualized in UCSF Chimera[44]. The nucleotide-free and nucleotide-bound conformations of LolCDE were aligned based on the TMD using the Matchmaker tool in Chimera. Distances between Cα atoms to measure conformational change-induced displacement was measured using the Structure analysis (distances) tool in Chimera. To measure the angle of rotation for the periplasmic domains of LolC a 2D plane was generated using the residues Leu256 and Met175 (Hook) in the nucleotide-free and nucleotide-bound conformations, using the Structure analysis (axes/planes/centroids) tool. The angle between the selected planes was measured using the *angle* command. A similar procedure was used to measure the rotation of the periplasmic domain of LolE using the amino acid residues Tyr260 and Leu110. Hydrophobicity of surfaces was measured using the *rangecolor* command. All figures and Movies were generated using Chimera.

**Reporting summary**. Further information on research design is available in the Nature Research Reporting Summary linked to this article.

## Data availability
The data that support this study are available from the corresponding author upon reasonable request. The three-dimensional cryo-EM density maps of *E. coli* LolCDE in nanodiscs have been deposited in the Electron Microscopy Data Bank under accession numbers: EMD-23783 (nucleotide-free) and EMD-23784 (vanadate-trapped). Atomic coordinates for the models of LolCDE have been deposited in the Protein Data Bank under accession numbers: 7MDX [https://doi.org/10.2210/pdb7MDX/pdb] (nucleotide-free) and 7MDY [https://doi.org/10.2210/pdb7MDY/pdb] (vanadate-trapped). Source data are provided with this paper.

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

## Acknowledgements

We thank L. Huang, X. Wang, and Z. Jiang from Westlake University cryo-EM facility for help in cryo-EM data acquisition for vanadate-trapped LolCDE. The dataset of nucleotide-free LolCDE was collected at cryo-EM core facility at the University of Massachusetts Medical School (UMMS). We are grateful to all Liao lab members for their helpful feedback throughout this project and A. Plummer and A. Culbertson for helpful comments on the manuscript. Y.L. was supported by the Zhejiang Provincial Natural Science Foundation of China (LR20C050001).

## Author contributions

M.L. conceived the project. M.L. and Y.L. supervised the project. S.S. and R.Z. performed molecular cloning, protein purification, nanodisc reconstitution, mutagenesis and co-purification. S.S., R.Z., and L.W collected the cryo-EM data. K.S. and C.X. helped cryo-EM data acquisition. S.S processed cryo-EM data and built models. S.F. performed mass spectrometry experiments. All authors contributed to data analysis. S.S., Y.L., and M.L. wrote the manuscript with input from all authors.

## Competing interests

The authors declare no competing interests.
