## [Peer Review File · Nature Communications]

Reviewers' Comments:

Reviewer #1:

Remarks to the Author:

The manuscript of Sharma et al. describes the cryo-EM structures of the ABC transporter LolCDE from gram negative bacteria (*E. Coli*) in the presence of a lipid-anchored lipoprotein in its nucleotide-free state and in the nucleotide-bound state at a resolution of 3.8 and 3.5 Angstroms embedded in nanodiscs. This transporter functions in the translocation of lipoproteins from the inner membrane to the periplasm on route to the outer membrane. The structures provide insight into how LolCDE recognizes the N-terminal triacylated peptide and lipid components of the lipoprotein and conformational changes that occur on nucleotide binding. The conformational changes involve movement of transmembrane segments and other domains leading to the collapse of the lipoprotein binding site and the extrusion of the lipoprotein from the inner membrane into the PD. The authors suggest that the mechanism is similar to that for MacB and classify LolCDE as a type VII ABC transporter.

Overall this is an interesting study by a group of investigators with considerable experience in structural biology of membrane proteins and in particular ABC transporters. The overall resolution is sufficient to obtain realistic structures and provide novel insight into how the N-terminal region of lipoproteins are recognized by LolCDE and apparently extruded from its binding site upon nucleotide binding. A few site-directed mutagenesis studies have been described in support of the binding of a lipoprotein Lpp to LolCDE using qualitative co-immunoprecipitation. However, these studies do not provide information on the effect of these mutations or Lpp binding on the ATPase activity or the extrusion function of LolCDE. This is a weakness of the paper. Nonetheless the studies are important in further defining how lipoproteins are processed by gram negative bacteria for transport to the outer membrane and adds to our general knowledge of the structure and transport mechanisms of ABC transporters.

Minor Issues to be Addressed:

1. Analysis of the kinetics of ATP hydrolysis: The kinetics of ATPase hydrolysis are carried out for LolCDE in nanodiscs. Presumably this is for LolCDE in the absence of lipoprotein since the latter is not added to the system for recycling the enzyme reaction. The authors should comment on whether the addition of substrate (lipoprotein) would enhance the ATPase activity of LolCDE in detergent or nanodiscs or if this is simply ATPase activity uncoupled to transport.
2. Site-directed mutagenesis studies appear to be directed only at the direct interaction of LolCDE with Lpp. Do the authors have any data to indicate how the loss of Lpp interaction with LolCDE due to mutations in LolCDE affect its ATPase activity. In the case where Lpp binding is lost due to mutations in Lpp, are endogenous lipoprotein bound to LolCDE as suggested in structural studies. Some clarification of this would help readers to further evaluate these site-directed mutagenesis studies.
3. Fig1b: Clarification: Does this graph correspond to a single experiment in which 3 measurements of ATPase activity were determined or three individual experiments with 3 measurements for each condition. The latter is more rigorous since activities can vary between experiments more so than measurements within a single experiment.
4. For figure 1e, it would be useful to define C, CH, and EH rather than have to wade through the manuscript to determine these abbreviations.

Reviewer #2:

Remarks to the Author:

In this manuscript, the authors solved two cryo-EM structures of ABC transporter LolCDE in nucleotide-free and nucleotide-bound states, respectively. Biochemical and mutagenesis studies were performed to analyze how LolCDE recognizes its substrate lipoprotein. Moreover, structural comparisons revealed the drastic conformational changes during the extrusion of the captured

lipoprotein. Generally, the manuscript is well written and of high quality. However, a recent paper (Nat Struct Mol Biol. 2021 Mar 29. doi: 10.1038/s41594-021-00573-x. Structural basis for bacterial lipoprotein relocation by the transporter LolCDE) have solved five cryo-EM structures of Escherichia coli LolCDE respectively in apo, lipoprotein-bound, LolA-bound, ADP-bound and AMP-PNP-bound states at a resolution from 3.2 to 3.8 Å, covering most states in the complete lipoprotein transport cycle. They have also performed a series of in vivo and in vitro activity assays, which provided insights into the sorting and transport of outer-membrane lipoproteins. This NSMB paper provided more functional data to support the structural insights into the lipoprotein recognition and transport mechanism of LolCDE. Given the major findings of the present manuscript have already been published elsewhere, it is not convincing to get published in Nature Communications. In addition, the authors should critically revisit the novelties of this work, cite the findings of the previous work and carefully revise the manuscript.

Major points :

1. Lines 151-152, "three acyl chains of lipoproteins occupy two hydrophobic pockets in LolCDE with the last attached acyl chain (R3) making the most extensive interactions". It seems that LolCDE have different preference to recognize the three acyl chains of the substrate lipoprotein. It would be helpful to individually test the binding affinities of LolCDE towards R1, R2, or R3. Moreover, site-directed mutagenesis together with binding assays should be performed to confirm the binding pocket of the lipoprotein.
2. Lines 165-166, "Because Lpp contains a serine residue in the +2 position, we modeled the first 6 residues of the lipoprotein as Cys-Ser-Ala-Ala-Ala-Ala". In fact, different types of lipoproteins in bacteria share a consensus motif of Cys-Ser at the N-terminus. It is more convincing to identify the substrate lipoprotein from the purified protein samples by SDS-PAGE and mass spectrometry.
3. Lines 176-178, "These are consistent with the notion that LolCDE recognizes highly variable lipoproteins through specific interactions with only their N-terminal peptide and lipid moieties". This hypothesis is the basis for the proposed model of substrate translocation by LolCDE. However, it lacks any experimental evidence to validate this hypothesis. Investigations on the interactions between LolCDE and Lpp mutants should be performed to support this conclusion.
4. Lines 180-181, "seems to suggest that lipoproteins enter the transporter from the front side through the interface between the TM1 of LolC and the TM2 of LolE". This is just a hypothesis. I would recommend the authors to introduce point mutations that form disulfide bond crosslink to block the entrance of lipoprotein. The in vitro transport assays should be performed to test this hypothesis.

Minor points:

1. lines 184-185: The wording of this sentence is improper, "Changing these two residues individually to alanine or lysine residue completely blocked Lpp co-purification"; It should be re-phrased "Changing these two residues individually to alanine or lysine residue completely blocked the attachment of Lpp to LolC and LolE"
2. Fig. 2. Please show the electron density for three acyl chains attached to the N-terminal peptide.
3. Fig. 2b was not cited in the manuscript.

Reviewer #3:

Remarks to the Author:

Lipoproteins are ubiquitous in bacteria. In Gram-negative bacteria, outer membrane targeted lipoproteins play essential roles in bacterial cell biology and pathogenesis. Lipoprotein transport is a highly conserved pathway. The first step in transporting lipoproteins to the outer membrane requires the essential LolCDE ABC transporter. The importance of LolCDE is underscored by recent efforts to

identify small molecules inhibitors as leads in antibiotic development.

The manuscript by Sharma et al. captures two key states of the lipoprotein transporter complex LolCDE. Thoughtful analysis and description of these structures illuminates key aspects of lipoprotein recognition by LolCDE and offers plausible molecular models for the extraction and transfer (to the periplasmic chaperone LolA) reactions.

By the nature of this work, the study is largely descriptive but a few experiments support novel insights revealed by the current structures. The data and analysis is pleasingly congruent with earlier studies of the system. The authors offer an explanation for why the N-terminal acyl modification of lipoproteins seems important for LolCDE interaction (this modification can be inactivated in vivo only if LolCDE is in excess). The authors also confirm and offer molecular details to support the earlier model that ATP-binding is the power stroke of the transporter. Finally, the authors are able to model an earlier interaction between LolA and a small periplasmic loop of LolC into both their structures.

The findings here offer important insights and the quality of data and presentation is excellent.

Comments:

1. The biggest advance here is a new mechanistic model for how lipoproteins inside the LolCDE transporter may be re-positioned to allow for transfer to the LolA carrier and a rest of the transport cycle. The authors propose an interaction between the N-terminal residues of lipoproteins and LolCDE (a recent LolCDE cryo-EM structure only considered the acyl chains). Their proposed model suggests that these interactions allow LolCDE to reorient the lipoprotein to facilitate movement through LolC and loading of LolA.

Direct evidence for the importance of LolE residues would strengthen the current model. At the very least, the authors should discuss how these LolE-lipoprotein interactions may occur since there is little conservation of residues in lipoproteins at sites +4 onwards.

2. The authors remove the C-terminal Lys58 from Lpp to prevent cell wall attachment of this lipoprotein which is toxic. However, the authors then attach a C-terminal Flag tag which itself terminates with a Lys. Do the authors see toxicity when expressing LppK58-Flag (ie. is some portion of their model lipoprotein unavailable due to cell wall attachment)? If LppF58-Flag is in fact attaching to cell wall, then data in 2j and 2k may need more clarification. It could be possible that LolCDE mutations cause inefficient cycling, allowing a significant population of LppF58-Flag to crosslink to cell wall and become unavailable for LolCDE interactions.

3. I think it would be helpful to readers (especially those in the pharmaceutical field) to add some discussion of known LolCDE inhibitors (pyrazoles from AstraZenica and Genentech) and the resistant mutations published previously. These mutations occur at high frequency and their likely effects have been poorly understood. The current structures may offer new insights and be useful for future drug developments aimed at LolCDE.

4. Discussion of the recent LolCDE data in NSMB is essential in revision.

RESPONSE TO REVIEWER COMMENTS

We would like to thank all reviewers for their constructive comments and suggestions. Our point-by-point response is presented below. The changes in the manuscript are highlighted in yellow or gray.

Reviewer #1:

The manuscript of Sharma et al. describes the cryo-EM structures of the ABC transporter LolCDE from gram negative bacteria (*E. Coli*) in the presence of a lipid-anchored lipoprotein in its nucleotide-free state and in the nucleotide-bound state at a resolution of 3.8 and 3.5 Angstroms embedded in nanodiscs. This transporter functions in the translocation of lipoproteins from the inner membrane to the periplasm on route to the outer membrane. The structures provide insight into how LolCDE recognizes the N-terminal triacylated peptide and lipid components of the lipoprotein and conformational changes that occur on nucleotide binding. The conformational changes involve movement of transmembrane segments and other domains leading to the collapse of the lipoprotein binding site and the extrusion of the lipoprotein from the inner membrane into the PD. The authors suggest that the mechanism is similar to that for MacB and classify LolCDE as a type VII ABC transporter.

Overall this is an interesting study by a group of investigators with considerable experience in structural biology of membrane proteins and in particular ABC transporters. The overall resolution is sufficient to obtain realistic structures and provide novel insight into how the N-terminal region of lipoproteins are recognized by LolCDE and apparently extruded from its binding site upon nucleotide binding. A few site-directed mutagenesis studies have been described in support of the binding of a lipoprotein Lpp to LolCDE using qualitative co-immunoprecipitation. However, these studies do not provide information on the effect of these mutations or Lpp binding on the ATPase activity or the extrusion function of LolCDE. This is a weakness of the paper. Nonetheless the studies are important in further defining how lipoproteins are processed by gram negative bacteria for transport to the outer membrane and adds to our general knowledge of the structure and transport mechanisms of ABC transporters.

RESPONSE: We thank the reviewer for the support and highlighting our key findings. We fully agree that examining the effect of the mutations or Lpp binding on the ATPase activity or the extrusion function of LolCDE is important. Our previous work on co-IP of lipoprotein with LolCDE has clearly demonstrated the effect of mutations on lipoprotein binding, which is a critical aspect of transporter function. To obtain further insights into the effects of mutations on ATPase activity, we have purified all mutants and test their activities of hydrolyzing ATP. These new results are presented in Supplementary Fig. 4i, j and described in lines 153-157 and 197-199. In general, the effects of mutations on Lpp binding and ATPase activity are not directly correlated.

Minor Issues to be Addressed:

1. Analysis of the kinetics of ATP hydrolysis: The kinetics of ATPase hydrolysis are carried out for LolCDE in nanodiscs. Presumably this is for LolCDE in the absence of lipoprotein since the latter is not added to the system for recycling the enzyme reaction. The authors should comment on whether the addition of substrate (lipoprotein) would enhance the ATPase activity of LolCDE in detergent or nanodiscs or if this is simply ATPase activity uncoupled to transport.

RESPONSE: Yes, the ATPase activities from our assay are for LolCDE in the absence of lipoprotein. Thanks for the suggestion. We have added a comment in lines 93-94: “*Because no lipoprotein was added in ATPase assay, the measured activities reflect the basal ATPase activity of LolCDE without active lipoprotein transport.*”

Following the reviewer’s suggestion, we measured the ATPase activity of purified LolCDE in DDM, with and without addition of purified Lpp. Our result showed no difference (see the figure below). This seems to suggest no effect on ATPase activity from Lpp interaction, but LolCDE may not bind or continuously transport Lpp. Thus,

future studies are needed to establish a robust lipoprotein transport assay and to reveal the correlation between the ATPase and substrate transport activities of LolCDE.

2. Site-directed mutagenesis studies appear to be directed only at the direct interaction of LolCDE with Lpp. Do the authors have any data to indicate how the loss of Lpp interaction with LolCDE due to mutations in LolCDE affect its ATPase activity. In the case where Lpp binding is lost due to mutations in Lpp, are endogenous lipoprotein bound to LolCDE as suggested in structural studies. Some clarification of this would help readers to further evaluate these site-directed mutagenesis studies.

RESPONSE: As mentioned above, to obtain further insights into the effects of the mutations on ATPase activity, we have purified all mutants and test their activity of hydrolyzing ATP. These new results are presented in Supplementary Fig. 4i, j and described in lines 153-157 and 197-199.

All OM-targeted lipoproteins bind LolCDE for their transport. In the case where Lpp binding is lost due to mutations of Lpp, other endogenous OM-targeted lipoproteins will still bind LolCDE.

3. Fig1b: Clarification: Does this graph correspond to a single experiment in which 3 measurements of ATPase activity were determined or three individual experiments with 3 measurements for each condition. The latter is more rigorous since activities can vary between experiments more so than measurements within a single experiment.

RESPONSE: Thanks for pointing this out. Fig. 1b is a representative of three independent experiments and each experiment contains three measurements of ATPase activity. The measured activities are very similar from individual experiments using different LolCDE preps. We have added a clarification in lines 595-596: “*Shown is a representative of three independent experiments. Each point represents mean ± s.d. of three measurements in one experiment.*”.

4. For figure 1e, it would be useful to define C, CH, and EH rather than have to wade through the manuscript to determine these abbreviations.

RESPONSE: Thanks for the great suggestion. We have added the definition of C, CH and EH in the legend of Fig. 1e (lines: 601-602).

Reviewer #2:

In this manuscript, the authors solved two cryo-EM structures of ABC transporter LolCDE in nucleotide-free and nucleotide-bound states, respectively. Biochemical and mutagenesis studies were performed to analyze how LolCDE recognizes its substrate lipoprotein. Moreover, structural comparisons revealed the drastic conformational

changes during the extrusion of the captured lipoprotein. Generally, the manuscript is well written and of high quality.

However, a recent paper (Nat Struct Mol Biol. 2021 Mar 29. doi: 10.1038/s41594-021-00573-x. Structural basis for bacterial lipoprotein relocation by the transporter LolCDE) have solved five cryo-EM structures of Escherichia coli LolCDE respectively in apo, lipoprotein-bound, LolA-bound, ADP-bound and AMP-PNP-bound states at a resolution from 3.2 to 3.8??, covering most states in the complete lipoprotein transport cycle. They have also performed a series of in vivo and in vitro activity assays, which provided insights into the sorting and transport of outer-membrane lipoproteins. This NSMB paper provided more functional data to support the structural insights into the lipoprotein recognition and transport mechanism of LolCDE. Given the major findings of the present manuscript have already been published elsewhere, it is not convincing to get published in Nature Communications. In addition, the authors should critically revisit the novelties of this work, cite the findings of the previous work and carefully revise the manuscript.

RESPONSE: Thank the reviewer for summarizing our current work and comparing it with the recently published LolCDE structure paper in NSMB. According to the journal policy (<https://www.nature.com/articles/s41467-020-17817-x>), independent work with overlapping results should be evaluated on its own merits.

In the Discussion of our revised manuscript (lines 296-327), we have included a new paragraph for comparison of our results with those in the NSMB paper. As the reviewer pointed out, most results from these two studies are consistent. However, several key differences in transporter activity and structures are apparent and important. Most differences likely stem from the fact that we used nanodisc-embedded LolCDE and NSMB work used detergent solubilized LolCDE.

Major points:

1. Lines 151-152, “three acyl chains of lipoproteins occupy two hydrophobic pockets in LolCDE with the last attached acyl chain (R3) making the most extensive interactions”. It seems that LolCDE have different preference to recognize the three acyl chains of the substrate lipoprotein. It would be helpful to individually test the binding affinities of LolCDE towards R1, R2, or R3. Moreover, site-directed mutagenesis together with binding assays should be performed to confirm the binding pocket of the lipoprotein.

RESPONSE: LolCDE prefers lipoproteins with all three acyl chains. R3 is added in the last enzymatic step before LolCDE interaction (see Fig. 1a). The observed acyl chain-transporter interaction is in the context of triacylated lipoprotein. For example, R3 occupies the back pocket together with R2. Therefore, testing LolCDE binding to lipoproteins with single acyl chains cannot recapitulate lipoprotein-LolCDE interaction. However, we agree with the reviewer that testing different binding pockets that interact with different elements of lipoproteins can provides insights of their importance for LolCDE binding.

As shown in Fig. 2j and 2k, we have performed site-directed mutagenesis and binding assay to test the important amino acid residues in the lipoprotein binding pockets. These results indicate critical importance of the Lole shoulder loop and back pocket in lipoprotein interaction. Since these regions are mainly involved in the interaction with R3 (and partially R2), our data are consistent with the notion of R3 being a critical element for LolCDE binding.

2. Lines 165-166, “Because Lpp contains a serine residue in the +2 position, we modeled the first 6 residues of the lipoprotein as Cys-Ser-Ala-Ala-Ala-Ala”. In fact, different types of lipoproteins in bacteria share a consensus motif of Cys-Ser at the N-terminus. It is more convincing to identify the substrate lipoprotein from the purified protein samples by SDS-PAGE and mass spectrometry.

RESPONSE: Thank the review for great suggestion. We have performed mass spectrometry experiment on purified LolCDE and identified various OM lipoproteins including Lpp (new Supplementary Fig. 1d and its source data).

As the reviewer suggested, we also modified the text and mention that many OM lipoproteins contain Ser in the +2 position (lines 172-173).

3. Lines 176-178, "These are consistent with the notion that LolCDE recognizes highly variable lipoproteins through specific interactions with only their N-terminal peptide and lipid moieties". This hypothesis is the basis for the proposed model of substrate translocation by LolCDE. However, it lacks any experimental evidence to validate this hypothesis. Investigations on the interactions between LolCDE and Lpp mutants should be performed to support this conclusion.

RESPONSE: In *E. coli*, the OM-targeted lipoproteins contain highly variable sequences. Their consistent structural features are the N-terminal cysteine residues with three acyl chains, as well as certain amino acid residues in +2 and +3 positions. These have been extensively studied (e.g., using many lipoprotein mutants) and are well established (see two reviews below).

1. Tokuda H, Matsuyama S. *Sorting of lipoproteins to the outer membrane in E. coli. Biochimica et biophysica acta* 1694, IN1-9 (2004).

2. Okuda S, Tokuda H. *Lipoprotein sorting in bacteria. Annu Rev Microbiol* 65, 239-259 (2011).

While a lot had been done to understand the importance of the residues in lipoproteins, little was known on the transporter side. Based on our structural findings, we have performed mutagenesis and binding assay (see Fig. 2j and k) to directly demonstrate the importance of specific regions in LolCDE for binding of the N-terminal peptide and acyl chains of lipoproteins.

4. Lines 180-181, "seems to suggest that lipoproteins enter the transporter from the front side through the interface between the TM1 of LolC and the TM2 of LolE". This is just a hypothesis. I would recommend the authors to introduce point mutations that form disulfide bond crosslink to block the entrance of lipoprotein. The in vitro transport assays should be performed to test this hypothesis.

RESPONSE: As the reviewer pointed out, this is currently a hypothesis. Determining the detailed pathway of lipoprotein translocation through LolCDE is beyond the scope of current study. We thank the reviewer's suggestion of using disulfide bond crosslinking to block lipoprotein entry and will perform this experiment in our future studies.

Minor points:

1. lines 184-185: The wording of this sentence is improper, "Changing these two residues individually to alanine or lysine residue completely blocked Lpp co-purification"; It should be re-phrased "Changing these two residues individually to alanine or lysine residue completely blocked the attachment of Lpp to LolC and LolE"

RESPONSE: We used "blocked Lpp co-purification" to directly describe the experimental finding. Our experiment cannot exclude the possibility of weakly attached Lpp being removed in co-IP experiment. However, we see the reviewer's point and have revised this part of sentence to "blocked stable attachment of Lpp to LolCDE" (line 196).

2. Fig. 2. Please show the electron density for three acyl chains attached to the N-terminal peptide.

RESPONSE: The electron density for three acyl chains attached to the N-terminal peptide is already shown in Fig. 2a-c and h, as well as in Supplementary Fig. 2g.

3. Fig. 2b was not cited in the manuscript.

RESPONSE: Thanks for catching that. We have now cited Fig. 2a-c in line 125.

Reviewer #3:

Lipoproteins are ubiquitous in bacteria. In Gram-negative bacteria, outer membrane targeted lipoproteins play essential roles in bacterial cell biology and pathogenesis. Lipoprotein transport is a highly conserved pathway. The first step in transporting lipoproteins to the outer membrane requires the essential LolCDE ABC transporter. The importance of LolCDE is underscored by recent efforts to identify small molecule inhibitors as leads in antibiotic development.

The manuscript by Sharma et al. captures two key states of the lipoprotein transporter complex LolCDE. Thoughtful analysis and description of these structures illuminates key aspects of lipoprotein recognition by LolCDE and offers plausible molecular models for the extraction and transfer (to the periplasmic chaperone LolA) reactions.

By the nature of this work, the study is largely descriptive but a few experiments support novel insights revealed by the current structures. The data and analysis is pleasingly congruent with earlier studies of the system. The authors offer an explanation for why the N-terminal acyl modification of lipoproteins seems important for LolCDE interaction (this modification can be inactivated *in vivo* only if LolCDE is in excess). The authors also confirm and offer molecular details to support the earlier model that ATP-binding is the power stroke of the transporter. Finally, the authors are able to model an earlier interaction between LolA and a small periplasmic loop of LolC into both their structures.

The findings here offer important insights and the quality of data and presentation is excellent.

RESPONSE: We thank the reviewer for the support and positive comments on our work.

Comments:

1. The biggest advance here is a new mechanistic model for how lipoproteins inside the LolCDE transporter may be re-positioned to allow for transfer to the LolA carrier and a rest of the transport cycle. The authors propose an interaction between the N-terminal residues of lipoproteins and LolE (a recent LolCDE cryo-EM structure only considered the acyl chains). Their proposed model suggests that these interactions allow LolE to reorient the lipoprotein to facilitate movement through LolC and loading of LolA.

Direct evidence for the importance of LolE residues would strengthen the current model. At the very least, the authors should discuss how these LolE-lipoprotein interactions may occur since there is little conservation of residues in lipoproteins at sites +4 onwards.

RESPONSE: We agree with the reviewer. Our structural studies revealed that the N-terminal sequence extends upward reaching the periplasmic domain of LolE, and that this domain predominantly rotates upon vanadate trapping. Thus, upon ATP binding/hydrolysis, LolE periplasmic domain likely pulls on the associated N-terminal lipoprotein sequence. Due to low sequence conservation (as the reviewer pointed out) and the requirement of establishing a sensitive lipoprotein transport assay, detailed analysis of the residues involved in the interaction between lipoprotein and the periplasmic domain of LolE is beyond the scope of current study. Interestingly, a recent study published on bioRxiv (<https://doi.org/10.1101/2021.01.05.425367>) analyzed the sequences of OM lipoproteins from *E. coli* and found that half of them contain a long and disordered N-terminal sequence; the length and disordered character of the N-terminal sequence of lipoproteins are required for optimal transport by LolCDE.

These results are consistent with our structural findings. We have added this reference and related discussion in lines 183-186.

2. The authors remove the C-terminal Lys58 from Lpp to prevent cell wall attachment of this lipoprotein which is toxic. However, the authors then attach a C-terminal Flag tag which itself terminates with a Lys. Do the authors see toxicity when expressing LppK58-Flag (ie. is some portion of their model lipoprotein unavailable due to cell wall attachment)? If LppF58-Flag is in fact attaching to cell wall, then data in 2j and 2k may need more clarification. It could be possible that LolCE mutations cause inefficient cycling, allowing a significant population of LppF58-Flag to crosslink to cell wall and become unavailable for LolCDE interactions.

RESPONSE: Thank the reviewer for pointing this out. We did not see any toxicity effect when expressing LppK58del-Flag. To avoid C-terminal lysine residue, we changed the tag to Myc tag (EQKLISEEDL) in all mutants and repeated the co-IP assay. The results (new Fig. 2j, k) are similar to those in the previous assay using Flag tagged LppK58del.

3. I think it would be helpful to readers (especially those in the pharmaceutical field) to add some discussion of known LolCDE inhibitors (pyrazoles from AstraZenica and Genetech) and the resistant mutations published previously. These mutations occur at high frequency and their likely effects have been poorly understood. The current structures may offer new insights and be useful for future drug developments aimed at LolCDE.

RESPONSE: This is a great suggestion. We have added discussion of known LolCDE inhibitors in lines 287-295.

4. Discussion of the recent LolCDE data in NSMB is essential in revision.

RESPONSE: Thanks for pointing this out. We have added a paragraph in Discussion (lines 296-327).

Reviewers' Comments:

Reviewer #1:

Remarks to the Author:

In the revised manuscript by Sharma et al. on the structure of Cryo-EM structures of LolCDE, the authors have provided additional data on the mutations described in their initial version and have addressed the concerns raised by the reviews.

The manuscript provides important, although partial, insight into the mechanism of transport of bacterial lipoproteins and therefore is of interest to readers of Nature Comm.

Reviewer #3:

Remarks to the Author:

The revised version of this manuscript addressed all my concerns. I have no further suggestions for this manuscript.